# Intravascular Large B-Cell Lymphoma: Two Cases Observed at a Single Institution

Rosanna Maria Miccolis [1,*], Gaetano De Santis [1], Caterina Buquicchio [1], Teresa Maria Santeramo [1], Mariangela Leo [1], Candida Rosaria Germano [1], Giovanna Lerario [1], Vera Carluccio [1], Sonia Mallano [1], Lina Cardisciani [2], Luisa Di Sciascio [2] and Giuseppe Tarantini [1,*]

1   UOC di Ematologia con Trapianto, Ospedale Monsignor R. Dimiccoli, 76121 Barletta, Italy
2   UOC di Emolinfopatologia School of Anatomic Pathology, Department of Biomedical and Neuromotor Sciences, University of Bologna, 40138 Bologna, Italy
*   Correspondence: rosanna.miccolis@aslbat.it (R.M.M.); giuseppetarantini0@gmail.com (G.T.)

**Abstract:** Intravascular large B-cell Lymphoma (IVLBCL) is a rare subtype of extranodal non-Hodgkin's lymphoma that is challenging to diagnose and has a poor prognosis. Here, we describe two patients newly diagnosed with IVLBCL at our institution: an African man with hemophagocytic-syndrome-associated IVLBCL and an Italian woman with a cutaneous variant of IVLBCL. They presented with very different clinical manifestations. Both cases were diagnosed in a timely manner, successfully treated, and achieved long-lasting remissions.

**Keywords:** extranodal lymphoma; blood vessels; autologous transplantation

## 1. Introduction

Intravascular large B-cell lymphoma (IVLBCL) is a rare type of extranodal large B-cell lymphoma with an almost exclusive localization of tumor cells within the lumen of blood vessels, mainly in smaller vessels and capillaries [1]. Because of the rarity of this disease, most of our knowledge on IVLBCL is derived from case reports and case series. Until recently, the relative paucity of cases has limited epidemiologic research; however, it is currently recognized as a disease primarily of the elderly, with a median age in the seventh decade of life, without sex prevalence [2]. Diagnosis is challenging, as clinical presentation is extremely heterogeneous. The disease can involve any organ system, causing a wide range of signs and symptoms ranging from less symptomatic forms presenting with fever, pain, or local symptoms to very aggressive variants combining B symptoms with rapidly progressing manifestations leading to multiorgan failure [3,4]. Historically, according to the clinical features and geographic predominance, two main subtypes of IVLBCL were recognized: the 'Western variant' and the 'Asian variant' [5]. The 'Western variant' is characterized by neurological symptoms (e.g., cognitive impairment, seizures, neuropathy, and paresis) and skin involvement (e.g., cellulitis, plaques, erythematous eruption, ulcerated nodules, and telangiectasia). The other type, the 'Eastern' or 'Asian' variant, is characterized by hemophagocytic syndrome (HPS), clinically manifesting as fever, hepatomegaly, splenomegaly, anemia, thrombocytopenia, and bone marrow involvement. In contrast with the Western variant, neurological symptoms and cutaneous lesions are rare. Considering recent findings and advances in knowledge, the recently updated World Health Organization (WHO) classification has suggested that the consideration of IVLBCL variants according to their clinical features rather than their geographical distribution is more appropriate. Based upon its clinical presentation, IVLBCL is separated into three variants [2,6]. The classical variant is the most common in Western countries, and the median age of patients is 70 years, without sex prevalence. The spectrum of clinical presentation is heterogeneous and ranges from paucisymptomatic forms, manifesting with a fever of unknown origin, pain, or organ-specific local symptoms, to a combination of B symptoms and signs of

multiorgan failure. Rapid deterioration in performance status (PS) is frequent. The skin and central nervous system (CNS) are the most frequently involved sites, and lymph nodes are usually spared [2,3]. The HPS-associated variant was historically known as the 'Asian' variant because it was almost without exception reported in Asian countries. Patients with this variant display a typical clinical hemophagocytic syndrome, represented by bone marrow involvement, fever, hepatosplenomegaly, and thrombocytopenia. These distinctive clinical features are often accompanied by nonneoplastic hemophagocytic histiocytes in peripheral blood or bone marrow smears. This variant is characterized by rapid aggressive onset and progression with a dismal prognosis [2,7]. The cutaneous variant presents with disease limited to the skin without any other site of involvement. It has been reported much more frequently in female patients at a median age of 60 years. Systemic symptoms could be present, but disease progression is less aggressive. Patients with the cutaneous variant seem to have a better prognosis and a longer survival period than those diagnosed with the classical variant [2,5,6]. Here, we describe two cases of IVLBCL observed at our institution.

## 2. Case Report

### 2.1. Case 1

A 43-year-old African man, who was an immigrant in Italy from Morocco, was admitted to the emergency department of our hospital after experiencing a 5-day 39 °C fever, sweats, fatigue, and abdominal pain. Moderate hypoxemia was noted. He was active and functional before symptom onset. He had no family history of malignancies and no known Asian family ancestry. Blood tests showed hemoglobin (Hb) 7.3 g/dL, white blood count (WBC) 10,720/mL, platelets (PLT) 49,000/mL, lactic dehydrogenase (LDH) serum level 5.852 U/L, and Beta2microglobulin 6.27 mg/L; microbiological tests were negative, including stool, urine, and blood cultures and serology for Epstein–Barr virus and cytomegalovirus. The anemia workup was negative for hemolysis, while a high ferritin level was revealed. The physical exam did not reveal skin rashes, lymphadenopathies, neurological deficits, palpable masses, or visceromegalies. A whole-body computer tomography (CT) scan was performed and showed modest bilateral pleural effusion and lung 'ground-glass' images, moderate hepatomegaly, and a significant splenomegaly (longitudinal diameter 17 cm). From the peripheral blood cytology, some 'Gumprecht nuclear shadows' and 7% immature lymphoid cells were detected. The combination of thrombocytopenia, hepatosplenomegaly, and fever in this clinical case led us to consider hemophagocytic syndrome, even though classic cytological features (i.e., the phagocytosis of nucleated blood cells) were lacking. Fluorodeoxyglucose positron emission tomography (FDG-PET) documented increased FDG uptake in the spleen only. A bone marrow (BM) biopsy and aspiration were performed: the flow cytometry showed a B-cell population positive for CD5, CD19, FMC7, and Lambda and negative for CD23. The BM histology (Figure 1) revealed a hypercellular marrow with myelodysplastic-like features of the myeloid compartment and a diffuse sinusoid engulfment by large pleomorphic lymphocytes, with one or more nucleoli. The immunohistochemistry study revealed that these cells were positive for CD20 (clone L26), CD10 (clone SP67), Bcl2 (clone SP66), Bcl6 (clone GI191E/A8), c-Myc (clone SP33), IRF4/MUM1 (clone IRF4/EP190); partially reactive for CD5 (clone SP19); and negative for CD3 (clone 2GV6), Cyclin D1 (clone SP4R), CD34 (clone Q-BEend/10), CD68 (clone PGM1), CD117 (clone YR145), and Myeloperoxidase (clone Myeloperoxidase). The purely intrasinusoidal growth was clearly highlighted by the immunohistochemistry study, which showed that cells were almost exclusively present within the vessel lumen (Figure 1), and a double staining for the CD34 and Pax5 antigens confirmed this feature. The infiltrate occupied approximately 15–20% of the whole marrow cellularity (evaluation conducted throughout the whole biopsy length); the in situ hybridization analysis (ISH) for EBER1/2 probe was negative. FISH studies were not performed.

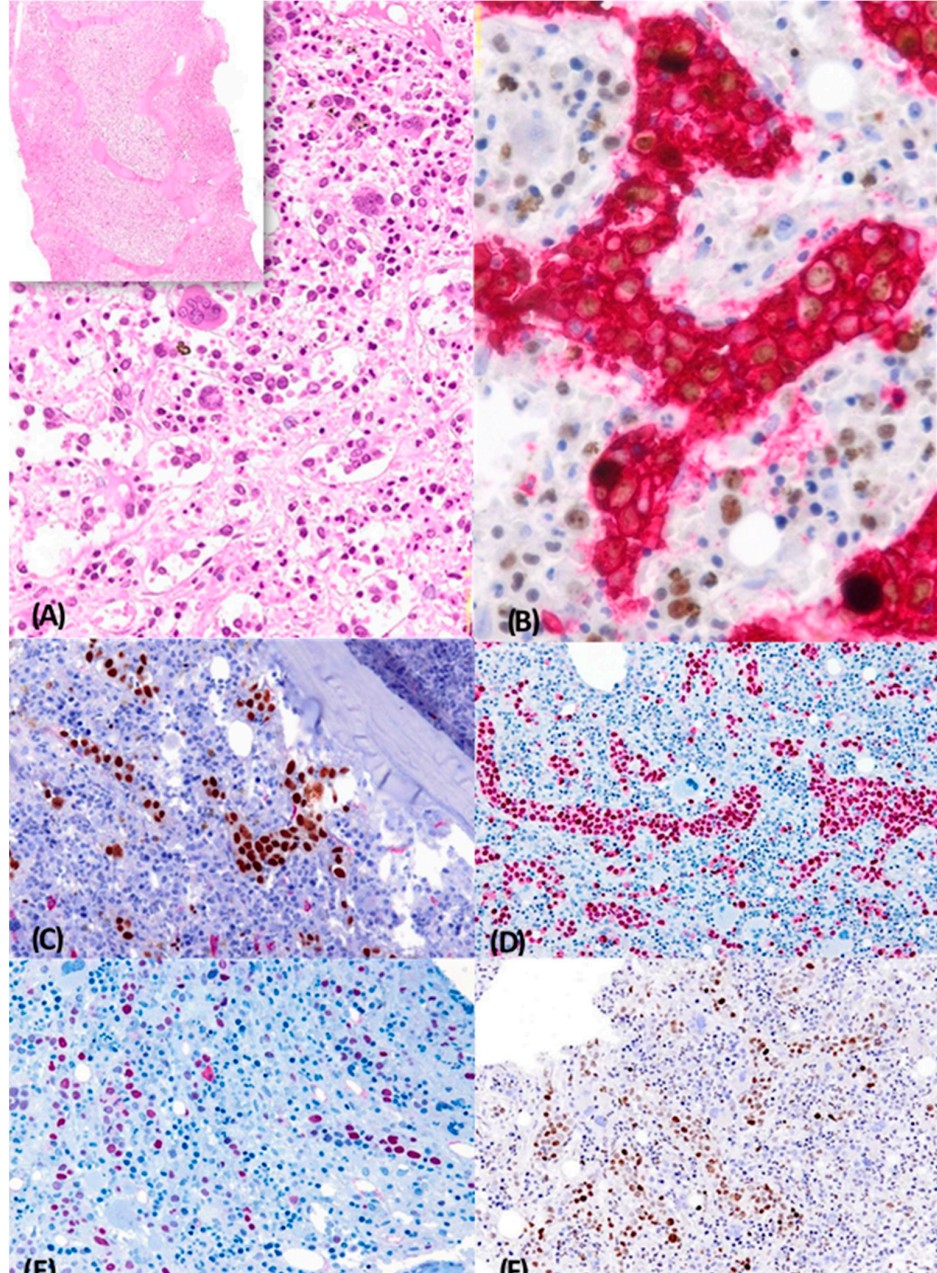

**Figure 1.** (**A**) H&E: the bone marrow biopsy showed high cellularity (inset; ×5), and at higher power magnification features of granulopoetic delayed maturation (MDS-like) and abnormal megakaryocytes were observed, as well as the presence of large, atypical cells within the vessel lumen (×20). (**B**) CD20/Ki67: the double staining for CD20 and Ki67 showed that the intrasinusoidal pleomorphic cells co-expressed CD20 and Ki67 (×40). (**C**) PAX5/CD34: the double staining highlighted that the PAX5-positive neoplastic cells were engulfed within blood vessels that were also stained focally positive for CD34 (×20). (**D**) IRF4: intense nuclear staining for MUM1/IRF4 antigen in the neoplastic cells; confirmation of the purely intravascular infiltration of the neoplastic cells is also provided by this image (×20). (**E**) c-Myc: nuclear staining for c-Myc protein in most neoplastic cells at moderate to strong intensity (×20). (**F**) Bcl6: nuclear staining for Bcl6 antigen at moderate to strong intensity (×20); confirmation of the purely intravascular infiltration of the neoplastic cells is also provided by this image (×20).

These results led to a diagnosis of intravascular large B-cell lymphoma stage IV "B", classified as high risk according to R-IPI. The chosen chemotherapy regimen was Rituximab plus dose-adjusted etoposide-doxorubicin-cyclophosphamide-vincristine-prednisone (DA-EPOCH) plus CNS prophylaxis with intrathecal liposomal cytarabine. After three cycles, a CT scan revealed the disappearance of pleural effusion and ground-glass images in the lungs and the complete resolution of hepatomegaly and splenomegaly. The fever, hypoxemia, and abdominal pain were resolved, and the biochemical profile was in a normal range, showing a LDH serum level of 420 U/L and a Beta2microglobulin level of 2.25 mg/L. A bone marrow biopsy performed after the fourth course of treatment showed no evidence of disease, and the FDG-PET documented a complete metabolic response. In consideration of the poor prognosis of the disease, we decided to consolidate these excellent results with autologous stem cell transplantation (ASCT). Cyclophosphamide (4 g/sqm) followed by granulocyte-colony-stimulating factor (GCS-F) were administrated as peripheral blood stem cell mobilization therapy, and CD34+ cell collection was then performed. The patient was referred to ASCT with Carmustine-Etoposide-Cytarabine-Melphalan (BEAM) conditioning therapy without unexpected non-hematological toxicity or engraftment difficulties, and he was discharged after 23 days of hospitalization. The patient remains relapse-free at four years of follow-up.

### 2.2. Case 2

A 76-year-old Italian (Caucasian) woman without a past medical history presented to our clinic with a two-month history of intermittent fever and fatigue. Mild normocytic anemia without leukocytosis or thrombocytopenia was present. She also complained of painful erythematous plaques on her left thigh and abdomen. Laboratory evaluations revealed: Hb 9.4 gr/dL with a mean corpuscular volume of 88 fl and reticulocytes of 1%; WBC count $5.4 \times 109/L$ (neutrophils 70%, lymphocytes 21%, and monocytes 8%) and platelet count $180 \times 109/L$; serum C-reactive protein (CRP) at 7.9 mg/dl, erythroid sedimentation rate of 41 mm/h, and serum LDH slightly elevated (552 UI/L). A lambda-type serum monoclonal gammopathy IgM was also present in a very small quantity (0.21 g/dL). A skin biopsy was performed which showed a subtle cell infiltrate made of large, atypical cells exclusively located within blood vessels (Figure 2). The immunohistochemistry study showed positivity for CD20 (clone L26), Bcl2 (clone SP66), Bcl6 (clone GI191E/A8), c-Myc (clone SP33), IRF4/MUM1 (clone IRF4/EP190); partial reactivity for CD5 (clone SP19); and negativity for CD3 (clone 2GV6) and CD10 (clone SP67). ISH assay for EBER1/2 probe was negative for EBV integration. FISH analysis was not performed. These findings were consistent with the diagnosis of IVLBCL. A bone marrow biopsy examination with immunohistochemistry and flow cytometry showed normal cellularity without neoplastic cells. The whole-body CT scan was unremarkable except for mild splenomegaly (longitudinal diameter 18 cm). An FDG-PET scan was also performed and showed a hypermetabolic enlarged spleen. The patient was started on combination chemotherapy with rituximab, prednisone, vincristine, cyclophosphamide, and doxorubicin (R-CHOP). A reduction in the number and size of cutaneous lesions was observed after the first course of chemotherapy. Cutaneous manifestations were completely undetectable after four courses. After completing six courses of chemotherapy, an FDG-PET scan documented complete metabolic remission, a CT scan demonstrated a reduction in splenomegaly (longitudinal diameter 13 cm), and the blood cell count was in a normal range. Three years after completing chemotherapy, there is no evidence of recurring disease.

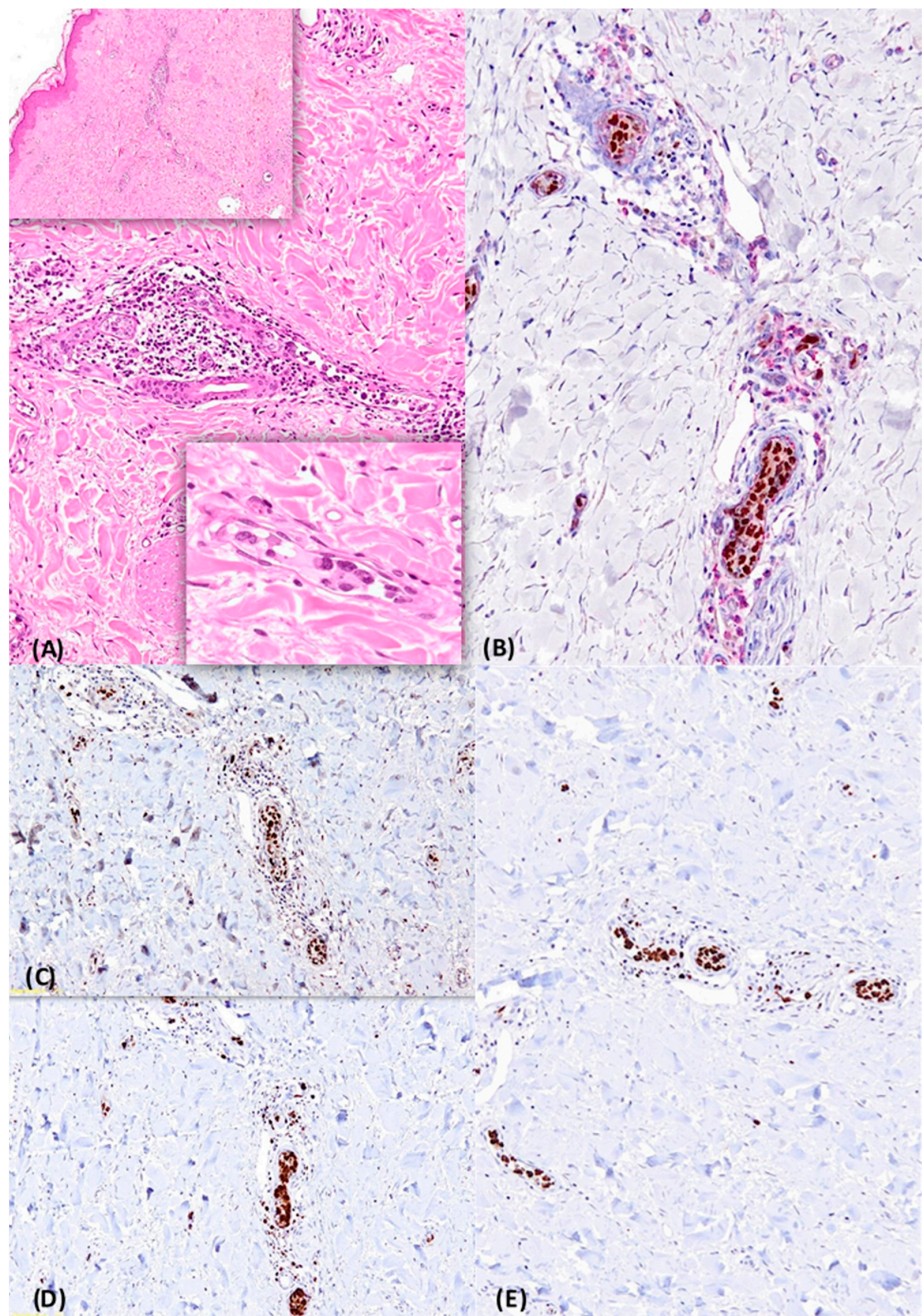

**Figure 2.** (**A**) H&E: the skin did not show significant cell infiltration in the dermis (inset; ×5), which could conversely be detected at a higher power (×20), being composed of large pleomorphic cells exclusively located within the vessel lumen (inset ×40). (**B**) CD31/PAX5: the double staining highlighted that the Pax5-positive neoplastic cells were only located within the lumen of blood vessels that were also stained positive for CD31 (×20). (**C**) Bcl6: nuclear staining for Bcl6 antigen at strong intensity (×20); confirmation of the purely intravascular infiltration of the neoplastic cells is also provided by this image. (**D**) IRF4: intense nuclear staining for MUM1/IRF4 antigen in the neoplastic cells; confirmation of the purely intravascular infiltration of the neoplastic cells is also provided by this image (×20). (**E**) Ki-67/MIB1: nuclear staining for Ki67 in most neoplastic cells; confirmation of the purely intravascular infiltration of the neoplastic cells is also provided by this image (×20).

The main clinical features and radiological and laboratory findings of the two cases are summarized in Table 1. The immunohistochemical and ISH studies are shown in Table 2.

**Table 1.** Clinical features of the two patients.

| Clinical Features | Case 1 | Case 2 |
|---|---|---|
| Sex/age (years) | M/43 | F/76 |
| Ethnicity | African | Caucasian |
| Primary location | BM | Skin |
| BM | Positive | Negative |
| Symptoms | Fever, fatigue, night sweats, abdominal pain, hypoxemia | Fever, fatigue |
| HB (g/mL) | 7.3 | 9.4 |
| WBC count | $10.7 \times 10^9/L$ | $5.4 \times 10^9/L$ |
| Platelets | $49 \times 10^9/L$ | $180 \times 10^9/L$ |
| LDH (U/L) | 5.852 | 552 |
| CRP (mg/L) | 20 | 7.9 |
| Albumin (g/L) | 1.9 | 3.8 |
| Ferritin (ng/mL) | 1120 | 450 |
| PTT (23.4–35.7 s) | 38 | 26 |
| PT (12–14.2 s) | 17 | 13 |
| CT | Hepatomegaly, splenomegaly, bilateral ground-glass opacities of lung and pleural effusion | Mild splenomegaly |
| PET-CT | Spleen FDG uptake | Spleen FDG uptake |

M, male; F, female; BM, bone marrow involvement; HB, hemoglobin; LDH, lactate dehydrogenase; CRP, C-reactive protein; CT, computed tomography; PET, positron emission tomography; FDG, 18F-fluorodeoxyglucose.

**Table 2.** Immunohistochemistry and ISH studies for the two cases of IVLBCL.

| | Case 1 | Case 2 |
|---|---|---|
| CD20 | + | + |
| CD79a | + | + |
| MUM 1 | + | + |
| BCL2 | + | + |
| BCL6 | + | + |
| CD3 | − | − |
| CD5 | +/− | + |
| Cyclin D1 | − | − |
| CD10 | − | − |
| c MYC | + | + |
| EBER-ISH | − | − |

## 3. Discussion

We reported two cases of newly diagnosed IVLBCL in a single institution in Italy. Case 1 was a North African man within the Western Hemisphere presenting with all clinical findings typical of the HPS–associated variant, such as thrombocytopenia, hepatosplenomegaly, bone marrow involvement, and fever in the absence of histological evidence of hemophagocytosis. In a recent analysis of 182 patients from a global case series, the hemophagocytic variant accounted for nearly 30% of the total cases and was found to have a negative prognostic impact [8]. Several case reports have described the hemophagocytic variant within the Western Hemisphere (Table 3). However, the majority involved either immigrants from Eastern countries or patients with a known family history of Asian descent. Theories regarding the origin of these geographical differences remain disputed. More recent conjectures have included potential ethnic differences involved in the production of inflammatory markers such as sIL2R, which appears markedly increased in Asian-variant patients in comparison to Western-variant patients. Helminthic infections may also represent an environmental trigger. In contrast to other hemophagocytic-associated lymphomas, Epstein–Barr virus (EBV) has been reported in only a few IVLBCL cases and is not thought to be the causative agent [9].

**Table 3.** Cases of the hemophagocytic variant of IVLBCL described in the Western Hemisphere in Caucasian, Hispanic, and African patients.

| Case 1 | Asian-variant intravascular lymphoma in the African race [9] | *Rare Tumors* (2012) | Holly Geyer et al. |
|---|---|---|---|
| Cases 2 and 3 | Retrospective study of intravascular large B-cell lymphoma cases diagnosed in Quebec [10] | Medicine clinical case report (2017) | Vanessa Brunet et al. |
| Case 4 | Asian-variant intravascular large B-cell lymphoma [11] | *Proc (Bayl Univ Med Cent)* (2017) | Derrick W. Su et al. |
| Cases 5, 6, and 7 | Intravascular large B-cell lymphoma in Hispanics: a case series and literature review [12] | *Journal of Community Hospital Internal Medicine Perspectives* (2020) | Sonha Nguyena and Zahra Pakbaz |
| Case 8 | Intravascular Large B-cell Lymphoma Successfully Treated with Autologous Transplantation [13] | *Clinical Lymphoma, Myeloma and Leukemia* (2021) | Alice Morigi et al. |
| Case 9 | Intravascular Large B Cell Lymphoma Still a Diagnostic Dilemma [14] | *Journal of Community Hospital Internal Medicine Perspectives,* Volume 12, Issue 1, Article 9 (2022) | Abuzar A. Asif et al. |
| Case 10 | Hemophagocytic Syndrome-Associated Intravascular Large B-cell Lymphoma With Dialysis-Dependent End-Stage Renal Disease Treated With Autologous Stem Cell Transplantation Using a Modified TEAM Regimen [15] | *Cureus* (2022) | Kama K. et al. |

The diagnosis of IVLBCL is achieved by demonstrating the presence of large lymphoma cells within small-to-medium blood vessels of major organs, including the skin [1]. Because of the intravascular progression of the neoplastic lymphoid cells and the tendency to mimic other diseases, the accurate and definitive diagnosis of IVLBCL is considered difficult. The antemortem diagnosis of this lymphoma is challenging, because there are no pathognomonic features or markers of IVLBCL, and the disease can be rapidly progressive. In recent analysis, postmortem diagnosis accounted for almost 20% of the total cases described in the literature [8,16]. In our Case 1 patient, a lymphoproliferative disease was soon suspected, as immature lymphoid cells were observed in a peripheral blood smear; however, this is a rare finding. In fact, despite the disease's intravascular growth pattern, only 5% to 9% of IVLBCL patients have peripheral blood involvement [2]. This finding encouraged us perform a BM biopsy early in the diagnostic work-up, leading us to a timely diagnosis. Our Case 2 patient presented with cutaneous lesions, and so diagnosis resulted from the skin biopsy. The skin is both one of the most frequent sites involved in the classical variant and the only site of involvement in the cutaneous variant, so discriminating between these two varieties can be challenging. No relationship has been found between the type of skin lesion and prognosis or systemic involvement [6]. Skin manifestations can trigger random skin biopsies (RSBs), which have now become one of the common methods for diagnosing IVLBCL. An RSB of healthy skin, in highly suspicious cases, has also proven to be highly sensitive and has increased diagnostic yield [14,17,18]. Both cases in this study demonstrated typical clinicopathological and immunohistochemical features of IVLBCL (CD20+, CD79a+, MUM1+, and CD3−). Both cases expressed CD5, which is positive in less than a half of cases described in the literature and seems to have no prognostic relevance [8,16]. Notably, both cases were dual expressors for BCL-2 and c-Myc. Studies have shown that the simultaneous expression of C-MYC and BCL2 may be a useful prognostic indicator. Compared with non-dual expressors, C-MYC/BCL2 dual expressors have a significantly higher mortality rate [19]. Unfortunately, our cases lacked FISH studies for MYC, BCL2, and BCL6 translocation. All patients diagnosed with IVLBCL are considered disseminated at the time of diagnosis. R-CHOP plus CNS prophylaxis is considered the treatment of choice in these patients. CNS prophylaxis is strongly recommended, because the CNS recurrence rate at 3 years is still as high as 25% [20]. R-CHOP is widely used as

a first-line therapy, but with a PFS at 2 years of only 50–60%, its outcomes are inferior to those of non-intravascular diffuse large B-cell lymphoma [21]. We chose the CHOP-derived scheme R-DA-EPOCH due to the possibility of dose adjustment in a young patient with a very poor clinical prognosis. ASCT has been reported to result in more favorable outcomes in individual cases and small case series [22,23]. A letter to the editor of *Bone Marrow Transplantation* that described the EBMT experience of ASCT in IVLBCL was published in December 2016. The complete dataset of the final cohort for analysis included 11 patients. Seven patients received ASCT as part of the first-line treatment, three as part of a salvage treatment, and for one patient this information was not available. Disease status at the time of ASCT was CR in eight patients and PR in three patients; the most commonly employed regimen for high-dose therapy was BEAM. After a median follow-up of 51 months for surviving patients, eight patients were alive and free from progression [24]. This was the first case series of ASCT in Western patients with IVLBCL in the Rituximab era. These data suggest that early ASCT may improve the results of standard induction in IVLBCL in patients up to an age of 65–70 years.

Even though the cutaneous variant of IVLBCL is less aggressive, it should be treated in the same way as the other variants. Radiotherapy could be an option for elderly patients unfit for chemotherapy with a single cutaneous lesion or should be used for palliative purposes only. Our Case 2 patient underwent a Comprehensive Geriatric Assessment (CGA) [25] and was assessed to be fit for chemotherapy, so she was treated with a R-CHOP standard dose; as the patient was given the better prognosis of the cutaneous variant, and taking account of her age, ASCT was not considered.

## 4. Conclusions

We described a rare case of Asian-variant IVLBCL in an African man within the Western Hemisphere. This case is of particular interest because descriptions of Asian-variant IVLBCL in non-Asian populations are limited. No obvious mechanism has yet surfaced explaining this rare phenomenon. An aggressive front-line treatment approach including ASCT can be suggested in younger and fit patients with HPS and highly aggressive disease. We also described a case of the cutaneous variant of IVLBCL in an Italian woman for whom less aggressive immuno-chemotherapy was employed, with a successful response. These two cases presented in very different manners, reflecting the great heterogeneity of IVLBCL's clinical presentation. A timely diagnosis could allow effective treatment and long-lasting remissions for patients affected by IVLBCL.

**Author Contributions:** Conceptualization, R.M.M. and G.T.; investigation, G.D.S.; data curation, G.D.S., C.B., T.M.S., M.L., C.R.G., G.L., V.C., S.M., L.C. and L.D.S.; writing—original draft preparation, R.M.M.; writing—review and editing, G.T. All authors have read and agreed to the published version of the manuscript.

**Funding:** This research received no external funding.

**Informed Consent Statement:** Informed consent was obtained from all subjects involved in the study.

**Conflicts of Interest:** The authors declare no conflict of interest.

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
