# Peer review of "Intravascular Large B-Cell Lymphoma: Two Cases Observed at a Single Institution"

_2673-8430, doi:10.3390/biomed3010004_

Round 1

Reviewer 1 Report

The group Tarantini performed a good work explaining the high IVLBCL heterogeneity, where two patients have been treated with success but in different way. Nowadays the heterogeneity of many diseases is the cornerstone to study personalized medicine and then tailoring of medical treatment to the individual characteristics of each patient.

Kind Regards

Author Response

I thank the reviewer for giving us the opportunity to improve this report and apologize for the delay. With this revision the title has been changed to avoid overemphasis as two cases do not account for a "series", data presentation has been reorganized in order to be clearer and discussion has been modified after updated litearture search. 

Reviewer 2 Report

Please find comments attached. 

Author Response

I apologize for the delay and thank the reviewer for his comments that give us the opportunity to improve this report. Please see the attachment.

Round 2

Reviewer 2 Report

I thank the reviewers for making substantial changes to the paper. Minor suggestions that (i) images of the CT scans would be beneficial for the readers to see and (ii) in both figures, panels must be relabeled as (A), (B)..etc. and the legends must describe each figure panel as (A) = H&E, (B) = KI67 and so on and so forth followed by the legend description. 

Author Response

I thank the reviewer for these comments and suggestions. (i) Imagines of the CT scans are unfortunately unavailable. (ii) In both figures panels have been relabeled as suggested.